# Improving the mental health and mental health support available to adolescents with social care-experience via low-intensity life story work: a realist review protocol

Simon P Hammond ![ORCID],[1,2] Claire Duddy ![ORCID],[3] Ella Mickleburgh,[2] Rachel Hiller,[4] Elsbeth Neil,[5] Kevin Williams,[6] Luke Rodgers,[7] Jon Wilson,[2] Geoff Wong[3]

For numbered affiliations see end of article.

**Correspondence to**
Dr Simon P Hammond;
s.hammond@uea.ac.uk

## ABSTRACT

**Introduction** Adolescents are the fastest growing group entering social care and are most at risk of mental ill-health. Life Story Work (LSW) is an existing transdiagnostic intervention thought to improve the well-being and mental health of children and adolescents under the care of a local authority by assisting the processing of trauma. Yet LSW is poorly evidenced, lacks standardisation and focuses on younger children. LSW is also high-intensity, relying on specialist input over several months. Adolescent-focused low-intensity-LSW is a promising alternative. However, there is poor evidence on how LSW, let alone low-intensity-LSW should be delivered to adolescents. We aim to identify why, how, in what contexts, for whom and to what extent low-intensity-LSW interventions can be delivered to adolescents with care-experience.

**Methods and analysis** Undertaking a realist review, we will: (1) develop an initial programme theory (PrT) of adolescent-focused low-intensity-LSW by consulting with two key expert panels (care-experienced and professional stakeholders), and by searching the literature to identify existing relevant theories; (2) undertake a comprehensive literature search to identify secondary data to develop and refine our emerging PrT. Searches will be run between 12/2021-06/2022 in databases including MEDLINE, PsycINFO, ASSIA and relevant sources of grey literature; (3) select, extract and organise data; (4) synthesise evidence using a realist logic of analysis and undertake further iterative data searching and consultation with our expert panels; (5) write up and share the refined PrT with our expert panels for their final comments. From this process guidance will be developed to help improve the delivery of LSW to support the mental health needs of adolescents with care-experience.

**Ethics and dissemination** Ethical approval is not required. Dissemination will include input from expert panels. We will develop academic, practice and youth focused outputs targeting adolescents, their carers, social, healthcare, and educational professionals, academics, and policymakers.

**PROSPERO registration number** CRD42021279816.

### Strengths and limitations of this study

► This is the first realist review of adolescent-focused low-intensity Life Story Work (LSW) and will improve our understanding of how this intervention may work in different settings and for different groups of adolescents with social care experience.

► Our review includes contributions from two separate public patient involvement groups featuring young adults with care-experience and professionals as recipients and deliverers of LSW.

► The contribution of two contrasting PPI groups, with differing potential agendas, may create issues in consolidating our final programme theory.

► Our review may be limited by the richness and relevance of evidence available in the literature.

## INTRODUCTION AND BACKGROUND

There are over 90 000 children and adolescents under the care of UK local authorities.[1] Adolescents are the fastest growing age group entering care in England[1] and the scale of their mental health needs is extraordinary for a 'non-clinical' population.[2] This group is up to six times more likely than their peers in the general population to experience mental ill-health[3] and 3–4 times more likely to attempt suicide.[4] Despite this, evidence indicates that the mental health needs of adolescents with social care-experience are under-reported and undertreated.[5]

The lifetime economic burden associated with outcomes stemming from child maltreatment, a central experience of many adolescents with care-experience, is estimated to be between £150 and 300 billion.[6 7] This is more expensive than the combined economic burden of major medical illnesses.[6] Cost stems from the high lifetime use of social-care and health services and loss of productivity,

BMJ

including high rates of unemployment (eg, almost 40% of adolescents who are not in education, training and employment are care-experienced[6–8]). Finding ways to improve the mental health of adolescents with care-experience represents a clear health, social care and educational priority.

Interventions to improve the mental health of adolescents with care-experience do exist.[9] However, when framed by the hierarchy of evidence for therapeutic studies,[10 11] the majority are costly and viewed as having a 'low-quality' evidence-base.[9 12–14] Being unable to answer vital questions such as what interventions work best, how, for whom, over what period and at what cost,[12] makes the commissioning of services very difficult and increases crisis-based referrals.[5 13 15]

A promising alternative to begin to address the unmet mental needs of adolescents with care-experience is the provision of quicker access to low-intensity services delivered at scale.[4 16–20] Low-intensity interventions vary according to whether their delivery involves support from a healthcare professional (guided self-help) or not (non-facilitated self-help), as well as the mode (face-to-face and/or digital), duration and intensity of services provided.[21 22] Early intervention and the delivery of low-intensity interventions by non-specialists could offer effective and cost-effective processes to improve mental health.[23] Evidence also indicates that early mental health interventions are more cost-effective than crisis-based referrals,[24] reducing pressure on already stretched health and social care services and providing evidence-based approaches to the commissioners of services.

Hence, when seeking to develop and evaluate low-intensity mental health interventions for adolescents with care-experience to address what the National Institute for Health Care Excellence describes as an '…urgent research priority …',[12] the first step is to understand how and why existing interventions 'work' or not in differing contexts, for whom and to what extent. This involves developing an explicit programme theory (PrT), detailing the underlying assumptions about how an intervention is meant to work and what impacts are expected.[25] Developing this in-depth understanding is critical in the case of adolescents with care-experience as they are a heterogeneous group.[9 26] Many will have complex histories and needs, meaning it is unlikely that an intervention with a single focus will address all of these needs.[9] This indicates that a transdiagnostic 'complex' intervention composed of several interacting components,[27] capable of being delivered in a timely fashion and flexible enough to match the changing needs of the young person with care-experience may prove effective.

Life Story Work (LSW) is an existing transdiagnostic intervention thought to improve the well-being and mental health of children and adolescents under the care of a local authority by assisting the processing of trauma. It is promoted in social care as a standard part of the care all children and adolescents with care-experience should receive. It is flexible, broad in focus and widely used,

illustrated via legislation underpinning its usage.[28–31] LSW is grounded in assumptions that constructing a coherent narrative is important for processing trauma(s) and that integrating new or corrective information can reduce negative emotions related to trauma, transitions and loss.[32–42] Typical LSW components include a therapeutic alliance (relationship with a trusted adult(s) capable of facilitating positive mental health), certain behaviours (individual or group therapeutic activities), procedures (prompts to action) and products (materials or artefacts).

However, despite the use of LSW being widely reported by people with care-experience, carers and professionals as valuable,[43–53] relatively little is known about how it works and the extent to which it works, especially for adolescents with care-experience.

A 2006 systematic review by McKeown *et al* on LSW in health and social care concluded that LSW had potentially far-reaching benefits but an 'immature' evidence-base.[47] In a 2020 scoping review that examined the peer-reviewed empirical evidence for LSW, the authors concluded that despite LSW being a clear priority for all stakeholders, it lacked an accepted standard for delivery and robust implementation, and evidence of effectiveness and cost-effectiveness.[54] In reviewing the 17 included studies, the authors highlighted several weaknesses of the current evidence base.[54] These included assumptions of 'standard LSW' without clear standardisation protocols, conceptualisations of LSW that did not appreciate the longitudinal nature of care-experiences across the life course, age-related limitations in terms of how LSW was understood and a lack of opportunity for innovation in practice and delivery.[54] A more recent paper has also highlighted the need for a broader appreciation of the mechanisms through which delivery may occur.[55]

A further weakness in the existing evidence base is that it does not sufficiently inform the development of LSW interventions. As noted by Hammond *et al*, the potential of low-intensity standardised transdiagnostic LSW approaches targeting adolescents is appealing, yet: '…without better evidence on what works best, how, for whom, over what period and at what cost we cannot move forward …'.[54]

In summary, adolescents are the quickest growing age group entering UK social care.[1] Adolescents with care-experience are up to six times more likely than their peers in the general population to experience mental ill-health, with their mental health needs often remaining unmet with significant individual, societal and economic life-long consequences.[3–8] LSW is a widely accepted and currently used intervention which is assumed to promote mental health, but its' evidence-base is limited.[9 46 47 54 56] Conventional LSW interventions are costly,[43 45 48 52] tend to focus on younger children with care-experience[54–58] and, like other interventions in children's social care, lack focus on the longer-term impact and attributable outcomes.[59]

Adolescent-focused low-intensity LSW interventions have the potential to improve the mental health of

adolescents with care-experience.[55] However, there is a clear need for research capable of building theoretically rich explanations of how low-intensity adolescent-focused-LSW works. Critically, this needs to be undertaken in a way that is flexible enough to recognise the varying home circumstances in which adolescents experience social care. Theory-led research is important because it can deliver findings that are usable to service providers and transferable to the different settings and adolescents they work with.[60]

## METHODS AND ANALYSIS

The aim of the current research is to begin to address the unmet mental health needs of adolescents with care-experience by improving the evidence base for, and developing guidance to inform the delivery of adolescent-focused low-intensity-LSW interventions. We know there is robust evidence that constructing coherent narratives are important mechanisms for processing trauma memories. However, the specific use of this method (LSW) with this population (adolescents with care-experience) in these settings (social care) is poorly informed theoretically and empirically. We will begin to address these gaps by asking:

> How, why, to what extent, for whom and in what circumstances can low-intensity LSW interventions, or elements of LSW interventions, be delivered to improve important and relevant outcomes for adolescents with care experience with mental health and wellbeing needs?

This research question is operationalised into two main objectives:

1. Undertake a realist review, to develop and refine a realist PrT that explains how and why adolescent-focused low-intensity-LSW interventions (or elements of interventions) may or may not work for adolescents with care-experience and in what contexts.
2. Use the realist PrT to produce preliminary guidance on the nature of good practice when delivering adolescent-focused low-intensity-LSW to adolescents with care-experience and hence provide benefits for them, their carers and health, social care and educational professionals.

### Realist review

We will address the research question and objectives by conducting a realist review. This approach will enable the team to deal with the complexity inherent in this research question, by accounting for the changing contexts of adolescents with care-experience across different settings and services. The realist approach is flexible enough to allow for the inclusion of the existing literature on LSW, alongside evidence that can provide transferable explanations for how and why other low-intensity mental health intervention strategies 'work' (and do not work) for adolescents with care-experience. We will follow the current RAMESES quality and publication standards for realist reviews in this project.[61]

### Patient and public involvement

Patient and public involvement (PPI) has been central to the design of the study and will continue to be a central component of this review. The area of interest originated from lead author SPH's time as a residential social care worker and the practice frustrations he faced when trying to engage adolescents with care-experience in LSW.

The project developed with PPI from discussions with team members with lived care leadership and lived care-experience LR and KW and adolescents with care-experience and practitioners from across England who highlighted numerous barriers to high-intensity LSW approaches and the '… childlike …' resources used with adolescents.

PPI coapplicants were integral to the inclusion of young adults with care-experienced expert panel (known as the Care-experienced Content Expert Group (CCEG)) alongside our multidisciplinary expert panel (known as the Content Expert Group (CEG)) which is comprised professionals within the area. The CCEG and CEG will meet three times during the project and provide insight into areas where the published and grey literature is lacking.

Our CCEG comprising young adults with care-experience and CEG panel will meet as separate groups before being brought together to meet (face-to-face and/or remotely) and contribute to the research asynchronously after the meetings (via WhatsApp text and video messaging). The CCEG will review information and feed into the ongoing iteration of the PrT and lead on the youth-centred elements of its dissemination. In this way, we will ensure that any outputs are reflective of the requirements of relevant stakeholders, something unlikely to be achieved through the literature review alone.

### Study design

We will follow a five-step process to conduct the review.

#### Step 1: develop an initial PrT

We will develop an initial PrT, created through reading the documents we have found during exploratory searches undertaken while preparing this research project. We will develop the initial PrT through project team meetings, where we will discuss and debate what the initial PrT should be. We will then hold the first of our CCEG and CEG meetings, presenting our initial PrT for feedback and further refinement.

The purpose of this step is to locate any existing theories of why and how low-intensity LSW interventions work (or are thought to work), in what contexts they work, to what extent and for whom. From these documents, we will identify any relevant existing theories of low-intensity LSW interventions and, where needed, will use techniques such as citation tracking and snowballing to obtain more data. At this stage we will also make use of project team

knowledge and contacts to identify relevant additional sources of information.[62] These informal techniques are particularly useful since, as we have already established, we will need to search widely for theories of adolescent-focused low-intensity LSW interventions in the literature. Throughout this step, we will regularly discuss information gathered until the initial PrT of adolescent-focused low-intensity LSW is formed.

## Step 2: evidence search

Following the creation of our initial PrT, we will undertake a comprehensive search of the literature, seeking secondary data to develop and refine it. This will include a review of published and grey literature, including educational materials for professionals and/or carers produced by professional bodies.

We will design, pilot and refine our search strategy with the input of an experienced information specialist CD. Our search strategy for this review will aim to update and build on searches undertaken in June and July 2020 to inform a scoping review undertaken by members of our team.[54] We will run searches in multiple research databases including MEDLINE, PsycINFO, ASSIA and Social Care Online along with relevant sources of grey literature. Our search strategy will include a comprehensive set of terms to describe the population of interest (adolescents, young people, looked-after, care-experience) and the intervention (LSW). In addition, we will undertake additional searches to identify documents containing data about other low-intensity mental health intervention strategies for adolescents with care-experience. For full details of the search strategies for these searches see online supplemental file 1.

To maximise the inclusion of relevant material, we will employ complementary searching techniques as appropriate, including citation searching (snowballing) and searching for 'sibling' or 'kinship' papers associated with included documents.[63 64]

Additional searching may be undertaken in response to new information requirements identified, and until we have obtained sufficient data ('theoretical saturation') to conclude that our refined PrT is coherent and plausible.

## Step 3: selecting, extracting, and organising data

Documents will be selected using a three-step screening process. First, the lead reviewer will screen all potentially relevant documents retrieved by the search by title and abstract, against inclusion and exclusion criteria. Our initial inclusion criteria are outlined in table 1.

Second, the full text of documents that met the inclusion criteria in the initial screen will be obtained and screened against the inclusion and exclusion criteria. Third, those that fulfil inclusion criteria will be read in detail and our final decision on inclusion in the review will be based on the criteria of relevance (does a document contain data that can contribute to the development of the PrT?) and rigour (were the methods used to generate the data trustworthy and credible?).[65] To ensure

**Table 1** Summary of eligibility criteria for realist review of adolescent-focused low-intensity Life Story Work for adolescents with care-experience

| Inclusion criteria | Exclusion criteria |
|---|---|
| Population: young people who are under the care of a local authority, young people who are 'looked after' or care experienced or adopted young people, or their parents/carers | Research focused solely on parenting style, communicative openness in foster or adoptive families, contact with birth family members |
| Intervention: LSW, including all activities involving recording, exploring, eliciting accounts of a care experienced person's life or personal history, to have an impact on their understanding of themselves and their identify And/or Low-intensity interventions that aim to address a mental health or well-being need | |
| Document type/study design: any | |
| Other: English language only | |

LSW, Life Story Work.

consistency in the application of the inclusion criteria we will use a process we have used before[66] and a 10% random sample of documents will be screened in duplicate at each stage by another member of the project team. Any discrepancies will be resolved through wider project team discussions.

Where necessary we will use established quality appraisal tools to judge the rigour of the data in included documents. For example, we will do this when a document contributes a substantial amount of data to our PrT and hence it is important for us to be able to trust these data by assessing the rigour of the methods used to generate the data. Where there is uncertainty as to how to judge rigour, we will predominantly consider the relevance of the data. In other words, we will likely include any relevant data. We will take this approach as even data that is of questionable quality may still provide relevant information to inform PrT development. To ensure that our PrT continues to provide plausible explanations for adolescent-focused low-intensity LSW, we will use an additional process for 'quality control'. That is, we will judge the explanatory plausibility of the PrT using the criteria of consilience, simplicity and analogy.[67] Despite these measures, threats to the plausibility of our PrT may still occur, particularly if sections of it are based predominantly on data that we would judge (globally) to be of questionable 'rigour'. In such cases, we will be explicit in highlighting and reporting these elements as limitations in project reports and future publications.

Data from all relevant full text documents will be extracted using a suitably designed and piloted standardised data collection process. We anticipate key

characteristics of each included document will be extracted into an Excel spreadsheet, and that the full text of documents will be uploaded to NVivo (a qualitative data analysis software) so relevant data can be organised and coded. Coding will involve extracting relevant sections of text from included documents according to how this data can contribute to PrT development.

## Step 4: synthesising evidence

Data analysis will involve the use of a realist logic of analysis with the goal of using the data from the documents to further develop the initial PrT developed in step 1. Data coding will be deductive (informed by the initial PrT), inductive (coming from the data within included documents) and retroductive (where inferences are made about underlying causal processes or mechanisms). Drawing on previous work,[68] we will use a series of questions about the relevance and rigour of content within documents as part of our process of analysis, as set out below:

Relevance:
► Are sections of text within this document relevant to PrT development?

Rigour (judgements about trustworthiness):
► Are these data sufficiently trustworthy to warrant making changes to any aspect of the initial and emerging PrT?

Interpretation of meaning:
► If the section of text is relevant and trustworthy enough, do its contents provide data that may be interpreted as functioning as context, mechanism or outcome?

Interpretations and judgements about Context-Mechanism-Outcome-Configurations:
► For the data that has been interpreted as functioning as context, mechanism or outcome, which CMOC (partial or complete) does it belong to?
► Are there further data to inform this particular CMOC contained within this document or other documents? If so, which other documents?
► How does this particular CMOC relate to other CMOCs that have already been developed?

Interpretations and judgements about PrT:
► How does this (full or partial) CMOC relate to the PrT?
► Within this same document are there data which informs how the CMOC relates to the PrT? If not, are there data in other documents? Which ones?
► Considering this particular CMOC and any supporting data, does the PrT need to be changed?

Data to inform our interpretation of the relationships between contexts, mechanisms and outcomes will be sought not just within the same source, but across sources (eg, mechanisms inferred from one document could help to explain the way contexts influenced outcomes for an intervention in another). Synthesising data from different documents is often necessary to compile CMOCs, since not all parts of the configurations can be found in the same document.

During the review, we will move iteratively between the analysis of particular examples, refinement of the PrT, and further iterative data searching to test particular theories (where needed).

During this step, we will hold our second CCEG and CEG meetings to discuss the literature, and sense check the developing PrT. The PrT and a summary of the literature will be discussed with these groups who will be asked to comment on its resonance with their perspectives and its implications for preliminary guidance. Wherever possible, we will address any gaps in the theory that persist (eg, through additional literature searches).

## Step 5: finalising the PrT and drawing conclusions

Near the end of the review, the refined PrT will be written up and shared during the final CCEG and CEG meetings for final comments. We will seek input to ensure the outputs we produce (outlined in the later section) are useful to all stakeholders and disseminated across 'lay', professional and academic networks.

## ETHICS AND DISSEMINATION

Ethical approval is not required as the realist review is secondary research.

The main outputs of this research will be an evidence-based PrT of adolescent-focused low-intensity-LSW that will inform our preliminary guidance which can be used to optimise any pre-existing practice immediately. We will share our final PrT using text, summary tables, a logical model and where appropriate, youth focused information clips and/or infographics to summarise individual papers/reports and draw insights across papers/reports.

For academic, clinical, social care and educational audiences, we will produce peer-reviewed journal articles, including those detailing the process and findings of the realist review and establishing the requirements for effective adolescent-focused low-intensity LSW. For other professional audiences we will actively share our preliminary guidance on the nature of good practice when delivering adolescent-focused low-intensity-LSW. This will take the form of articles and blogs.

**Author affiliations**
[1]School of Education and Lifelong Learning, University of East Anglia, Norwich, UK
[2]Norfolk and Suffolk NHS Foundation Trust, Norwich, UK
[3]Nuffield Department of Primary Care Health Sciences, University of Oxford, Oxford, UK
[4]Division of Psychology and Language Sciences, University College London, London, UK
[5]School of Social Work, University of East Anglia, Norwich, UK
[6]The Fostering Network, London, UK
[7]The Care Leaders, Oxford, UK

**Contributors** SPH: conceptualisation, methodology, writing-original draft preparation, writing-review and editing, supervision, funding acquisition, project administration. CD: funding acquisition, conceptualisation, formal analysis, data

curation, original draft preparation, writing-review and editing, visualisation, investigation. EM: original draft preparation, conceptualisation, methodology, funding acquisition, writing-review and editing, project administration, visualisation, project administration, investigation. RH: funding acquisition, conceptualisation, writing-review and editing. EN: funding acquisition, conceptualisation, writing-review and editing. KW: funding acquisition, conceptualisation, writing-review and editing, resources. LR: funding acquisition, methodology, conceptualisation, writing-review and editing, resources. JW: funding acquisition, conceptualisation, writing-review and editing. GW: funding acquisition, conceptualisation, methodology, original draft preparation, writing-review and editing, supervision.

**Funding** This project is funded by the National Institute for Health Research (NIHR) Research for Patient Benefit Programme (Grant Reference Number NIHR 201963).

**Competing interests** SPH offers consultancy services in social media and Digital Life Story Work via https://www.digitallifestorywork.co.uk/. LR is Director of Strategy of The Care Leaders, KW is the CEO of The Fostering Network. The remaining authors declare that they have no known competing financial interests or personal relationships that could have appeared to influence the work reported in this paper.

**Patient consent for publication** Not applicable.

**Provenance and peer review** Not commissioned; externally peer reviewed.

**ORCID iDs**
Simon P Hammond http://orcid.org/0000-0002-0473-3610
Claire Duddy http://orcid.org/0000-0002-7083-6589

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
