## [Reviewer comments · BMJ Open]

ARTICLE DETAILS

TITLE (PROVISIONAL)	Improving the mental health and mental health support available to adolescents with social care-experience via low-intensity life story work: A realist review protocol
AUTHORS	Hammond, Simon; Duddy, Claire; Mickleburgh, Ella; Hiller, Rachel; Neil, Elsbeth; Williams, Kevin; Rodgers, Luke; Wilson, Jon; Wong, Geoff

VERSION 1 – REVIEW

REVIEWER	Duffy, Michael University of Ulster at Magee
REVIEW RETURNED	31-Dec-2021

GENERAL COMMENTS	This review is important for several of reasons: - adolescent mental health is a huge social problem and many young people in the social care system experience mental health problems that are undiagnosed and often untreated;- the use of LSW is used widely in social care settings, as a low intensity intervention, despite being poorly evidenced and lacking a standardised approach- there is robust evidence that constructing a coherent narrative is an important mechanism for processing trauma memories but the specific use of this method (LSW) with this population (adolescents) in these settings (social care) is poorly informed theoretically and empirically It is important that the authors are explicit in reporting the limitations of data that would be judged as questionable in terms of 'rigour' given the breadth of literature to be included It is also important that the research team have experience doing this type of research and reporting on findings from such reviews
---

REVIEWER	Poudel-Tandukar, Kalpana University of Massachusetts Amherst, Elaine Marieb College of Nursing
REVIEW RETURNED	04-Jan-2022

GENERAL COMMENTS	Thank you for the opportunity to review the paper. This interesting paper reported a planned study that aimed to develop low-intensity Life Story Work intervention that can be delivered to adolescents with care experience. The study objectives are novel; however, the paper needs to include a specific rationale, procedural details, and significance in addressing the stated objectives.
--

	Abstract: The dates of the study should be included in the abstract. Introduction: I would suggest the authors strengthen the justification for their study need to improve the current understanding of adolescents' mental health outcomes. For example, what novel information will their study add to the existing body of literature? What strategies do they plan to take to minimize current challenges? How will they work to enhance the quality of care of adolescents in improving the mental health outcomes? A specific description of the novelty and significance of the study will strengthen the quality of the paper. Methods: Authors have mentioned the search strategies, including search terms, search duration, inclusion criteria, and steps for program development. They also explained the assessment procedure of study quality and the risk of bias. It would be interesting to add some specific examples authors might expect after completing each step of program development to give readers some sense of tentative contents of the program outlook.
--	---

VERSION 1 – AUTHOR RESPONSE

Reviewer: 1

Dr. Michael Duffy, University of Ulster at Magee

Comments to the Author:

This review is important for several of reasons:

- adolescent mental health is a huge social problem and many young people in the social care system experience mental health problems that are undiagnosed and often untreated;
- the use of LSW is used widely in social care settings, as a low intensity intervention, despite being poorly evidenced and lacking a standardised approach
- there is robust evidence that constructing a coherent narrative is an important mechanism for processing trauma memories but the specific use of this method (LSW) with this population (adolescents) in these settings (social care) is poorly informed theoretically and empirically

- Author's response – thank you for your kind words in summarising the need for the study.

- It is important that the authors are explicit in reporting the limitations of data that would be judged as questionable in terms of 'rigour' given the breadth of literature to be included

- Author's response – We thank the reviewer for their comment. This was something we feel we had stated on pages 10 and 11 in the first submission but have amended this to make this more overt to readers.

- It is also important that the research team have experience doing this type of research and reporting on findings from such reviews

- Author's response – We thank the reviewer for their comment and are confident with the team we have assembled a strong team to enable us to implement and execute the research. We have content experts (SH, RH, EM, JW, BN) experts via experience (LR, KW) and realist review methodologists (CD and GW). we are aware of the RAMESES Publication standards for realist reviews and have

added to the manuscript on page 7 that we will be reporting the realist reviews using the existing RAMESES publication standards.

Reviewer: 2

Dr. Kalpana Poudel-Tandukar, University of Massachusetts Amherst

Comments to the Author:

Thank you for the opportunity to review the paper.

This interesting paper reported a planned study that aimed to develop low-intensity Life Story Work intervention that can be delivered to adolescents with care experience. The study objectives are novel; however, the paper needs to include a specific rationale, procedural details, and significance in addressing the stated objectives.

Abstract: The dates of the study should be included in the abstract.

- Author's response – We thank the reviewer for their comments.

Introduction: I would suggest the authors strengthen the justification for their study need to improve the current understanding of adolescents' mental health outcomes.

- Author's response – We thank the reviewer for their comments. We feel, as per reviewer 1's comments, that the justification for the study's need is clear and strong in its current form. However, we have included the following summary section prior to the methodology to ensure the individual and societal life course costs are brought to the fore to underline the urgency and significance of the study.

- o "In summary, adolescents are the quickest growing age group entering UK social care [1]. Adolescents with care-experience are up to six times more likely than their peers in the general population to experience mental ill-health, with their mental health needs often remaining unmet with significant individual, societal and economic life-long consequences [3-8]. LSW is a widely accepted and currently used intervention which is assumed to promote mental health, but its' evidence-base is limited [9, 46, 47, 54, 56]. Conventional LSW interventions are costly [43, 45, 48, 52], tend to focus on younger children with care-experience [54-58] and, like other interventions in children's social care, lack focus on the longer-term impact and attributable outcomes [59].

Adolescent-focused low-intensity LSW interventions have the potential to improve the mental health of adolescents with care-experience [55]. However, there is a clear need for research capable of building theoretically rich explanations of how low-intensity adolescent-focused-LSW works...." (pages 5-6)

For example, what novel information will their study add to the existing body of literature?

- Author's response – We thank the reviewer for their comments. We do believe that our study will add to the existing body of literature both theoretically and empirically. Our belief is shared by reviewer 1's comments. To provide clarity on how our project will provide novel findings, we have added the following text (inspired from reviewer 1's comments) to page 6:

- o "We know there is robust evidence that constructing coherent narratives are important mechanisms for processing trauma memories. However, the specific use of this method (LSW) with this population

(adolescents with care-experience) in these settings (social care) is poorly informed theoretically and empirically. We will begin to address these gaps by asking...” (page 6).

What strategies do they plan to take to minimize current challenges? How will they work to enhance the quality of care of adolescents in improving the mental health outcomes?

o Author’s response – We are unclear what the reviewer is referring by “current challenges” and would be grateful for clarity on this comment if required by the editor.

A specific description of the novelty and significance of the study will strengthen the quality of the paper.

o Author’s response – We are thankful for this comment, but we feel this is something already within the narrative of the paper.

Methods: Authors have mentioned the search strategies, including search terms, search duration, inclusion criteria, and steps for program development. They also explained the assessment procedure of study quality and the risk of bias. It would be interesting to add some specific examples authors might expect after completing each step of program development to give readers some sense of tentative contents of the program outlook.

o Author’s response – We thank the reviewer for their comments. We appreciate that it may be helpful to some readers that we provide examples of what we might find from each step of our realist review. However, we are reluctant to provide any findings at this stage as anything we provide would be speculative. We also wonder if this is something that would be expected for a protocol paper, which has the main purpose of enabling us to transparently report what we intend to do in our review (as opposed to provide any initial findings – no matter how speculative). We will be publishing our final findings and in it will provide the transparency needed for the reader to understand what the findings are from each step of our review.

Reviewer: 1

Competing interests of Reviewer: none

Reviewer: 2

Competing interests of Reviewer: I do not any competing interests.